# Modified Nucleosides, Nucleotides and Nucleic Acids via Click Azide-Alkyne Cycloaddition for Pharmacological Applications

**DOI:** 10.3390/molecules26113100

**Published:** 2021-05-22

**Authors:** Daniela Perrone, Elena Marchesi, Lorenzo Preti, Maria Luisa Navacchia

**Affiliations:** 1Department of Chemical, Pharmaceutical and Agricultural Sciences, University of Ferrara, 44121 Ferrara, Italy; mrclne@unife.it (E.M.); prtlnz@unife.it (L.P.); 2Institute of Organic Synthesis and Photoreactivity National Research Council, 40129 Bologna, Italy

**Keywords:** click chemistry, 1,2,3-trizole, azide-alkyne cycloaddition, bioisosteres, nucleosides, nucleic acid, oligonucleotides, pharmaceutical approach

## Abstract

The click azide = alkyne 1,3-dipolar cycloaddition (click chemistry) has become the approach of choice for bioconjugations in medicinal chemistry, providing facile reaction conditions amenable to both small and biological molecules. Many nucleoside analogs are known for their marked impact in cancer therapy and for the treatment of virus diseases and new targeted oligonucleotides have been developed for different purposes. The click chemistry allowing the tolerated union between units with a wide diversity of functional groups represents a robust means of designing new hybrid compounds with an extraordinary diversity of applications. This review provides an overview of the most recent works related to the use of click chemistry methodology in the field of nucleosides, nucleotides and nucleic acids for pharmacological applications.

## 1. Introduction

Nucleosides and their corresponding phosphate forms (nucleotides) are considered the building blocks of nucleic acids and are implicated in all biological functions of cellular life, i.e., metabolic regulation, catalysis and energy transfer. This broad and crucial role suggests that the chemistry of these molecular families represents a pivotal research topic in bioorganic and medicinal chemistry. The search for novel and efficient strategies to synthetize a large number of analogs has made “click chemistry” a widely used concept in the field of medicinal chemistry and drug discovery. The copper (I)-catalyzed Huisgen 1,3-dipolar cycloaddition of azides with terminal alkynes is probably one of the most applied click reactions thanks to its high specificity and efficiency in connecting two different molecular entities [1]. It was independently introduced by Sharpless and Meldal in 2001 to provide regioselectively 1,2,3-triazole moieties [2,3]. While the uncatalyzed Huisgen cycloaddition of alkynes and azides (AAC, Figure 1A) was performed at high temperatures and gave a mixture of 1,5-and 1,4-triazoles [4,5], the copper-catalyzed azide/alkyne cycloaddition (CuAAC, Figure 1B) provides only the 1,4-disubstituted 1,2,3-triazole regioisomer in mild conditions. It can be done in both organic solvents and aqueous conditions. The latter, introduced originally by Rostovtsev et al., 2002, are extremely commonly used in biochemical conjugations as well as in organic preparations, ensuring easy isolation and high purity of products [1]. In addition, it tolerates almost every functional group, accelerates the reaction time by seven orders of magnitude [2], has a very favorable atom economy [6] and the work-up procedures are often straightforward. The CuAAC reaction can be performed in both homogeneous and heterogeneous conditions and with a great variety of copper sources [7]. Most frequently, CuSO_4_ is used in combination with sodium ascorbate as a reducing agent in aqueous conditions; otherwise, the copper halides CuI and CuBr commonly used in organic solvents require the presence of an amine base and/or microwave or thermal heating [8] or ultrasonication [9] to accelerate the reaction rate and thus the formation of cycloaddition products. Nevertheless, certain drawbacks are associated with the use of Cu(I) in biological systems [10], including several unintended side reactions in cells (i.e., oxidation of histidine and arginine residues) [11] and DNA degradation [12], where long-term exposure is required. The presence of Cu-stabilizing ligands in CuAAC reaction limits cytotoxic effects [13] and has been recommended for DNA labeling [14]. Furthermore, the inclusion of 10% DMSO, a known radical scavenger, has proven to suppress oxidative damage without strongly inhibiting the CuAAC reaction [15]. By exchanging copper for ruthenium, 1,5-disubstituted 1,2,3-triazoles can also be regioselectively accessed as reported by Fokin and Jia in 2005 [16]. Even though this ruthenium-catalyzed azide–alkyne cycloaddition (RuAAC, Figure 1C) does not appear as widely diffused as CuAAC reactions, reports of its applications in medicinal chemistry and biochemistry are increasing quickly [17]. A more biofriendly method of activating alkyne toward reaction with azide is the strain-promoted azide–alkyne cycloaddition (SPAAC, Figure 1D) that proceeds under physiological conditions avoiding the use of transition metal catalysts [18]. The use of strained cyclooctyne reagents enables the reaction to proceed efficiently and with a comparable kinetic to the Cu-catalyzed reaction in live cells as well [19]. Nevertheless, the SPAAC approach lacks regiospecificity, forming two triazole regioisomers in approximately equal amounts.

Triazoles have a widespread occurrence in different bioactive compounds thanks to several interesting properties resulting in improved pharmacokinetic and pharmacological profiles of molecular entities [20,21,22], and several triazole-containing drugs are in clinical study in a wide range of medical indications [23]. Triazoles are stable to basic and acid hydrolysis and under reductive and oxidative conditions. Nevertheless, the triazole moiety is more than a passive linker, since the heterocycle shows several interesting biophysical properties and structural features, enabling it to act as a pharmacophore and bioisostere. Indeed, 1,2,3-triazoles readily associate with biological targets through hydrogen bond formation, dipole–dipole and π-stacking interactions and finally show a marked stability to metabolic degradation. Moreover, the structural features, such as rigidity of the ring, polarity and capability of hydrogen bonding, enable 1,2,3-triazoles to mimic different functional groups, although they do not occur in nature. This accounts in part for the poor knowledge of biological pathways and biocompatibility of 1,2,3-triazole moieties. Recently, pharmacophore and linker properties, and also the bioisosteric behavior of 1,2,3-triazole rings have been well documented, and novel classes of 1,2,3-triazole-containing hybrids, conjugates and analogs of natural compounds have been synthesized and evaluated as lead compounds for different biological targets [24]. It is reasonable to assume that an appealing synthetic approach such as “click azide-alkyne cycloaddition” has attracted extensive interest in drug discovery in general [25,26,27] and then in the synthesis of modified and/or conjugated nucleosides, nucleotides and nucleic acids, including oligonucleotides. The aim of the present review is not to be comprehensive, but rather to highlight the extraordinary diversity of pharmacological applications related to the use of click chemistry methodologies in the field of nucleosides, nucleotides and nucleic acid. We focus on articles published since 2015.

## 2. Clicked Nucleosides and Nucleotides

The relevance of click chemistry for nucleoside functionalization has pushed synthetic studies for the preparation of modified purine and pyrimidine nucleosides carrying azide- or alkyne-reactive handles on the sugar or the nucleobase moiety. A selection of alkyne- and azide-nucleoside derivatives commercially available and from recent literature are reported in Figure 2.

Thanks to their biological properties, as well as their different chemical/biochemical reactive moieties, nucleosides can be considered as an interesting platform for conjugation with bioactive/functional molecules. The design of hybrid molecules represents, for instance in cancer therapy, an interesting approach to enhance biological activity with respect to both cytotoxicity and cytoselectivity and to reduce multidrug resistance. Click chemistry, representing a very effective strategy for bioconjugation, has been applied for the preparation of nucleoside hybrids with the aim to discover new potential bioactive agents such as anticancer or antiviral.

A library of nucleoside-bile acid conjugates prepared by combining 2′-deoxyadenosine, 2′-deoxyguanosine and 2′-deoxyuridine, as well as adenosine and guanosine alkyne derivatives, with a selection of bile acid azides (cheno-, urso-, nor-cheno-, nor-urso- and taurourso-deoxycholic acid) by means of the click reaction was reported by Navacchia et al. [28]. The target conjugates were obtained in 60–90% yield by means of click reaction between alkyne-nucleosides **10**–**13** or commercially available alkyne-nucleoside **2** and the proper bile acid azide under commonly used conditions, that is, (1:1:1.5) H_2_O/t-BuOH/THF (*v*/*v*) in the presence of the CuSO_4_ catalyst and sodium ascorbate under stirring at room temperature for 18 h. Scheme 1 depicts the synthesis of adenine based nucleoside-bile acid conjugates **15a,b** as an example. The nucleoside-bile acid conjugates were tested in vitro against leukemic K562 and HCT116 colon carcinoma, as well as on normal fibroblast cells. Six compounds out of twenty-three displayed interesting anti-proliferative activity against the selected cancer lines and negligible cytotoxic effects against normal fibroblasts. 

In the framework of a study on nucleoside bioconjugates with a dual mode of action for cancer treatment, the same research group designed a novel 2′-deoxyadenosine derivative conjugated with a nitric oxide photodonor molecule. Hybrid **17** was obtained by reacting alkyne **12a** with 1-nitro-2-(trifluoromethyl)aniline azide **16** via click chemistry (Scheme 2) [29].

Compound **17** was tested against leukemic K562 and HCT116 colon carcinoma, as well as on normal fibroblast cells. Comparison of cytotoxic activity between hybrid **17** and that of the similar previously reported 2′-deoxyadenosine-based hybrid with an S-alkyl linker **S**-**dAdo** [30] instead of triazole, allowed estimating in the first instance that the presence of the triazole moiety could improve both cytotoxicity and cytoselectivity.

Ruddarraju et al. [31] reported the synthesis and biological evaluation for anticancer activity of a series of nucleoside-theophylline hybrids with 1,2,3-triazole linkage. The hybrids were prepared by click reaction mediated by Cu(I) between azide-nucleoside and acetylene theophylline derivatives. Scheme 3 depicts the synthesis of the most potent hybrids of the series **20** and **21** that displayed significant anticancer activity against all cancer cell lines tested such as lung (A549), colon (HT-29), breast (MCF-7) and melanoma (A375).

CuAAC click chemistry was employed by Kozarski et al. [32] for the preparation of a series of novel analogs of 7-methylguanosine 5′-monophosphate (m7GMP), a central metabolite of mRNA cap degradation. The compounds were evaluated as inhibitors of human 5′-nucleotidase cNIIIB. In particular, compound **24**, containing a 5′-1,2,3-triazoylphosphonate moiety (Scheme 4), was shown to have inhibitory activity against HscNIIIB together with sufficiently low affinity for eIF4E and DcpS, two major cytoplasmic proteins responsible for the recognition of m7G nucleotides in cell small-molecule inhibitor of HscNIIIB. Therefore, the 5′-1,2,3-triazoyl moiety was identified as a new structural motif that may be useful for the development of nucleotide-based inhibitors targeting 5′-nucleotidases.

Aminoribosyl uridines derivatives **25** and **26** were reacted with complementary azide and alkyne partners, respectively, via CuACC click chemistry to introduce a variety of long hydrophobic chains on the triazole moiety (Scheme 5). The new compounds were evaluated in vitro as inhibitors of the bacterial transferase MraY [33]. Among others, compounds **27a,b** and **28a,b** showed a highly improved inhibitory effect against MraY with respect to the starting alkyne **25** and azide **26** respectively (Scheme 5). 

Several triazolyl 13α-estrone-nucleoside (thymidine, 2′-deoxycitidine and 2′-deoxyadenosine) bioconjugates prepared by CuAAC reaction were also reported [34]. For instance, CuAAC conjugation reaction between 5′-azido-2′deoxyadenosine **29** and 3-*O*-propargyl-13α-estrone **30** was optimized to obtain the corresponding bioconjugate **31** in 68% yield by using CuI (Scheme 6). All compounds were tested for the antiproliferative activity against A2780, HeLa and MCF-7 cancer cell lines. In all cases, the cytotoxicity of bioconjugates was lower than that of the parent 13α-estrone.

A series of glycomimetics of UDP-GlcNAc containing a triazole ring in place of the β-phosphate unit were prepared by using CuAAC chemistry, in order to evaluate their affinities for human O-GlcNAc transferase (hOGT) [35]. Slightly different reaction conditions were employed for click reaction depending on the anomeric azide/triazole: the classical copper(II) sulfate/sodium ascorbate system performed well for α-anomers **35a**,**b**, while the reactions leading to β-anomers **34a**,**b** showed better results with the system (EtO)_3_P·CuI (Scheme 7). The target compounds β-**38a**,**b** and α-**39a**,**b** were obtained after coupling of the α and β glycosyltriazoles with the phosphoramidite **40** and removal of protecting groups. Their affinities for human O-GlcNAc transferase (hOGT) were evaluated by biological assays. *K*_i_ measurements with human OGT in vitro showed a moderate binding only for the β anomer **38b** (*K*_i_ = 231.9 μM).

Nucleoside analogs are also a rich source of antiviral agents. Recently, Wang et al. [36] reported the synthesis and biological evaluation for anti-HIV activity of betulinic acid-nucleoside hybrids by reacting a selection of azido pyrimidine ara-nucleosides **41a,b** with 2-propargyl betulinic acid **42** via click chemistry. Hybrids **43a,b** displayed highly potent anti-HIV activity (see IC_50_ inhibitory concentration values reported in Scheme 8) and significantly lower cytotoxicity compared to starting azido ara-nucleosides **41a,b** (see CC_50_ cytotoxic concentration values against MT4 normal cells reported in Scheme 8).

Click chemistry has also been employed for the synthesis of novel nucleosides analogs carrying the unnatural heterocycle triazole as nucleobase. Liu et al. [37] recently reported novel arabinofuranosyl triazole nucleoside analogs with anti-HVB (Hepatitis B Virus) activity. In particular, nucleoside analog **47** was obtained through a multistep synthesis including, as a key step, the click reaction mediated by CuI between azido **44** and alkyne **45** (Scheme 9), yielding the triazole derivative **46**. Compound **46** was further processed to obtain 1,2,3-triazole-2′-deoxy-2′-fluoro-4′-azido nucleoside analog **47**. Compound **47** displayed high activity against lamivudine-resistant HBV mutants. The antiviral activity of **47** was tested in HepG2.2.15 cells and confirmed by the inhibitory effects on HBeAg and HBsAg secretion after treatment. The results compared to that of Lamivudine, a drug approved by the FDA for the treatment of chronic hepatitis B, indicate that triazole **47** could actually be a candidate for the development of an alternative or complementary therapy for HVB infection (Scheme 9).

Sabat et al. [38] reported the synthesis of a ribofuranosyl-1,2,3-triazole obtained via Huisgen cycloaddition (Scheme 10). Compound **52** was succcessfully evaluated for its anticancer activity against a panel of nine cancer cell lines vs. C-nucleoside analog Ribavirin. Compound **52** displayed remarkably higher cytotoxicity than Ribarivin, especially against human breast adenocarcinoma (MDAMB231), human melanoma (MDAMB435) and prostatic carcinoma (DU145) cancer cells (Scheme 10).

Cobb et al. [39] reported the synthesis of a library of spirocyclic nucleoside analogs tested for antivirial activity. A series of azido-alkyne **53** were treated under thermal conditions in toluene for 24 hto obtain the corresponding spirocyclic nucleoside analogs **54** via intramolecular 1,3-dipolar cycloaddition (Scheme 11). All compounds were tested against MHV (Murine Hepatitis Virus) and some of them showed promising activity. In particular, compound **54** with R = 4-Cl-C_6_H_4_ (63% overall yield) was found to strongly reduce the virus growth in the range of 1–2 mM concentration with a dose-effectiveness profile similar to the antiviral nucleoside Ribavirin. 

In recent years, azido nucleosides have been also subjected to SPAAC reaction for the preparation of fluorescent nucleosides for living cells imaging. Zayas et al. reported [40] a protocol for convenient SPAAC of 2- or 8-azido adenine or 5-azido uracil-based nucleosides with various cyclooctynes and its application for imaging in living cancer cells MCF-7 by direct fluorescence light-up. For instance, the reactions between 8-azido adenosine **4** and cyclooctynes **55** and **57** led to hybrids **56** and **58,** respectively, in almost quantitative yield under mild conditions in both pure polar organic solvent and water mixture as reported in Scheme 12.

CuAAC and SPAAC click reactions have been recently reported for the preparation of analogs of the potent antiviral perylene 2′-deoxyuridine derivative dUY11 [41]. In particular, reaction of cyclooctynes **59** with 3′-*O*-azidomethyl-dUY11 **60** afforded the target triazole derivative **61** in 70% yield as a non-separable regioisomer mixture within 5 min (Scheme 13). Antiviral activity of compound **61** was evaluated against tick-bone encephalitis virus (TBEV). The introduction of the triazole derivative moiety led to a remarkably increased anti-TBEV activity of compound **61** with respect to that of parent dUY11 (EC50 = 0.0075 and 0.024 μM respectively) as well as increased solubility in aqueous DMSO. 

Very recently, Pogula et al. [42] reported on the synthesis of novel nucleosides with antibacterial activity via CuAAC click reaction. Briefly, azido uridine **62** was reacted with propargyl alcohol via CuSO_4_/ascorbate-mediated reaction to obtain the triazole derivative key intermediate **63** (Scheme 14). Subsequent functionalization of hydroxymethyl residue to aldehyde and then to alkyne moiety, followed by the coupling with 5-iodouracil led to compound **64** displaying highly potent and selective activity against Giardia trophozoite growth (Scheme 14).

5-Ethynyl pyrimidine and 8-ethynyl purine nucleosides have been recently reacted with TMSN_3_ via CuI or CuSO_4_/sodium ascorbate catalyzed cycloaddition to give chemoselectively the corresponding 5- and 8-triazolyl derivatives respectively [43]. The introduction of a simple triazole moiety resulted in good fluorescent properties. In particular, 8-triazolyl 2′-deoxyadenosine derivative **65** exhibited the highest quantum yield of 62% (Scheme 15). Compound **65** was successfully incorporated into a duplex DNA for fluorescent imaging purposes. 

Recently, Kondhare et al. [44] reported the preparation of clickable 5-aza-7-deaza 2‘-deoxyguanosine analogs with linear alkynyl or dendritic tripropargylamino moiety **66A-C** (Scheme 16). Subsequent click reaction mediated by CuSO_4_ with 1-azidomethylpyrene **67** led to fluorescent nucleoside analogs **68**–**70** applicable for diagnostic or imaging purposes (Scheme 16). 

## 3. Clicked Nucleic Acids

Nucleic acid therapeutics are a novel promising class of drugs owing to the fact that they are designed to be specific towards their targets [45,46]. As such, the possibility of developing drugs that target a single gene or RNA species instead of treating the proteins they are responsible for is thought to be a more attractive approach for various diseases. However, several issues, such as unfavorable cellular uptake, biodistribution, toxicity and size barrier (i.e., oligomers with more than one hundred bases) have limited the clinical development of these strategies [47,48]. Biorthogonal click chemistry represents a promising approach to address several challenges. Ju et al. were the first to report on the use of click chemistry to synthesize fluorescent modified oligodeoxyribonucleotides with high selectivity, yield and stability [49]. The original strategy was then extended in recent years to a wide range of applications in the field of nucleic acid research [50,51,52]. A detailed and critical discussion about the chemistry, the available modified nucleosides and applications of azide−alkyne cycloaddition reactions in nucleic acid chemistry was published very recently [53].

In this review, we considered dividing clicked nucleic acids for therapeutic applications in two main classes regarding the functional use of triazoles. The first consists of analogs of nucleic acids that possess the artificial triazole linkage as a biocompatible mimic of natural phosphate backbone as shown in Figure 3B [54,55]. Although at first glance the triazole moiety does not seem a structural analog of a phosphodiester linkage of DNA, its structural similarity was shown by Nuclear Magnetic Resonance (NMR) studies of a model DNA duplex containing an isolated triazole linkage in comparison with the unmodified counterpart [54]. Moreover, the compatibility of an unnatural triazole backbone linkage with DNA and RNA polymerase enzymes in vitro and in living organisms was described [56].

The second class includes “clicked” nucleic acids in which 1,2,3-triazole acts as a biocompatible linker to afford new hybrid entities with improved therapeutic potential.

### 3.1. Triazole-Llinked Nucleic Acid Analogues

In a pioneering study, Dondoni et al. reported a model study for the preparation of a triazole-linked trinucleoside of thymidine [57]. A solid-phase approach amenable to large-scale preparation of a 10-mer triazole-linked DNA by using copper-catalyzed Huisgen cycloaddition was soon after reported by Isobe et al. [58]. Interestingly, the ^TL^DNA formed a stable double strand with a natural complementary DNA strand, paving the way for a new class of oligonucleotide analogues with inertness toward nuclease hydrolysis. More recently, a series of chimeric 21-mer RNA oligonucleotides that possessed the triazole linker at different positions were synthesized using solution-phase click-chemistry and solid-phase automated synthesis (Scheme 17) [59]. Two different jointing units (**71** and **72**) were prepared and coupled in solution phase by a copper-catalyzed cycloaddition reaction to produce the triazole-linked dinucleotides **UtU**, **CtU** and **CtC** subsequently converted into the corresponding phosphoroamidites with moderate to good yields for the two steps (77–87%). Selected triazole-linked dinucleotides were then incorporated in a 21-mer oligonucleotide to produce a modified siRNA targeting a gene of the enhanced yellow-fluorescent protein (EYFP). Varying the position of triazole-linked dinucleotides, thirteen oligonucleotides were prepared using the automated phosphoramidite method without affecting the coupling efficiency. Finally, the chimeric siRNA oligonucleotides were examined for the silencing activity revealing the preference of the triazole modifications.

Aimed at developing therapeutic nucleic acids, Hayes et al. reported on the synthesis of triazole-linked morpholino (^TL^MO) oligonucleotides (Scheme 18) [60]. The triazole-linked morpholino hybrid dinucleotide **75** that combined the tested synthesis of ^TL^DNA with the high stability associated with morpholino chemistry was prepared from alkyne **73** containing a morpholine nucleoside and azido thymidine **74** via CuAAC reaction and conversion to the corresponding phosphoramidite. Hybrid **75** was then incorporated into a 13-mer sequence via solid phase synthesis. UV melting experiments were executed to evaluate thermal stabilities of the novel ^TL^MO-containing oligomer with annealed DNA and RNA. For the DNA binding, it was found that, although the T_m_ was lower than that of the control DNA:DNA duplex (T_m_ = 62.4 °C), the morpholino incorporation in the triazole-linked sequence improved the binding (T_m_ ^TL^MO:DNA = 56.1 °C) compared to the ^TL^DNA (T_m_ ^TL^DNA:DNA = 55.1 °C). The stabilizing effect of the morpholino moiety was more pronounced in the case of duplexes with RNA, where the ΔT_m_ between ^TL^MO:RNA (T_m_ = 56.6 °C) and the control DNA:RNA (T_m_ = 58.5 °C) was −1.9 °C. Instead, a −4.4 °C difference was determined between the control and ^TL^DNA:RNA (Tm = 54.1 °C) duplex. Nevertheless, for future therapeutic applications, further structural studies were required. 

Conformationally restricted locked nucleic acid (LNA) is a bicyclic RNA analog with interesting antisense properties [61,62]. At the same time, but independently, the research groups of Watts et al. [63], and Brown et al. [64] reported on the synthesis and biological properties of triazole-linked LNA. Envisaging that the presence of LNA around the triazole could improve the binding affinity of triazole-linked nucleic acids, several LNA-modified dinucleotides were synthesized by CuAAC chemistry and then incorporated into ONs and siRNAs following standard protocols. Studies of binding affinity showed that the triazole-linked ribo and xylo-LNA (**78A–C**, Scheme 19) could be incorporated, without compromising binding affinity, at the 3′ or 5′-termini of oligonucleotides, but not at internal positions [63]. On the other hand, internal incorporation of LNA on the 3′-side of the triazole linkage (**81**, Scheme 20A) by CuAAC in conditions previously described [65] significantly improved the thermal stability of DNA:RNA duplexes [64]. In contrast, duplex stabilization was not observed when LNA was incorporated on the 5′-side of the triazole (**84**, Scheme 20B). Finally, the combination of 5′-LNA-triazole-3′-LNA provided greater resistance to exonuclease digestion relative to LNA alone, suggesting that they should be highly resistant to enzymatic degradation in vivo.

CRISPR–Cas9-mediated genome engineering, considered an essential tool to treat or even cure genetic disorders, many forms of cancer, viral infections and other disorders, received the 2020 Nobel Prize in Chemistry [66]. As a consequence of its wide use, the demand for gRNA synthesis has substantially increased, and access to tailored gRNAs is limited. Enzymatic methods can be complex, time-consuming and difficult to scale up, while existing chemical routes are inefficient considering that 100-mer RNAs remain at the limit of solid-phase synthesis. Several efforts have been made to address these problems. Recently, Brown et al. reported an optimized split-and-click approach to modified sgRNAs in which an artificial triazole backbone linkage was incorporated by simple CuAAC chemistry [67]. The synthesis of sgRNAs was first split into a variable DNA/genome-targeting 20 mer **85** bearing an alkyne and a fixed Cas9-binding chemically modified 79-mer **86** bearing an azide. Click ligation of the two components, the second step of the synthesis, generated sgRNAs **87** containing an artificial triazole backbone that is well tolerated in functionally critical regions of the sgRNA enabling effective Cas9-mediated DNA cleavage in vitro and in cells (Scheme 21). Finally, this approach provided easier access to bespoke libraries of modified sgRNAs.

### 3.2. Nucleic Acid Conjugates via 1,2,3-Triazole Linkers

#### 3.2.1. Solid Phase Supported Click Conjugation

Synthetic advantages of click chemistry were also exploited for the attachment of several entities to solid-supported oligonucleotides. In a recent work, Honcharenko et al. [68] reported for the first time a method for the efficient conjugation of peptides to phosphorothioate oligonucleotides (PS-ON) (Scheme 22). The use of phosphorothioate as a backbone is a widely employed strategy to enhance the pharmacokinetics and nuclease resistance of oligonucleotides [69], and to date, the majority of therapeutic ONs incorporate the PS backbone modification [43]. However, there are few examples of the successful use of the CuAAC conjugation to oligonucleotides with a substantial PS content, probably due to cleavage and desulfurization of PS linkages, which can occur in the presence of catalytic amounts of CuI [70]. The method involved a post-synthetic derivatization of solid-supported PS oligonucleotides **88** with a linker to introduce an alkyne at the 5′-termini of the ON. Derivatized **89** was reacted with a blood–brain barrier penetrating peptide (MIF-1) [71] containing an azido moiety, by click chemistry using the CuBr·Me_2_S complex as a superior catalyst. Interestingly, it was found that the reaction time was a critical factor in the desulfurization process, and a total yield of 99% was achieved for **90** with a reaction time of 2 min. It could be pointed out that the method opens the way for further synthesis of therapeutically relevant PS-oligonucleotide conjugates for in vivo delivery.

Another recent study reported a simple solid-phase CuAAC procedure for the preparation of 5′-end single- or multi-labeled modified oligonucleotides for therapeutic and diagnostic applications [72]. The authors envisaged the use of multiple GalNac moieties as in vivo hepatocytes targeting residues. The fully automated synthetic strategy allowed diverse and easily scalable modifications and could, in the future, bring a better understanding of the correlation between GalNac residues spacing and asialoglycoprotein receptor interaction. The optimization of the Cu source (CuI·P(OEt)_3_) and conditions allowed the synthesis of sophisticated oligonucleotide-GalNAc dendrimer such as **92** and **93** conjugates showed in Scheme 23. Remarkably, two consecutive CuACC were carried out on fully protected oligonucleotides 91, which, after removal of protecting group and resin cleavage, afforded the multi-labeled oligonucleotides with minor retention of copper in the sample.

#### 3.2.2. Post-Synthetic-Click Conjugation

Antisense oligonucleotides and small interfering RNA (siRNA) are of great interest for their potential as therapeutics. Currently, there are two siRNA based drugs approved by the FDA and EMA: Patisiran, encapsulated in a lipid nanoparticle for delivery to hepatocytes [73], and the latest Givosiran [74], a siRNA covalently linked to a six *N*-acetylgalactosamine (GalNAc) ligand to enable specific delivery to hepatocytes. In fact, in vivo delivery of siRNAs is the key challenge to enable their use for therapeutic approaches. Recently, a study on the conjugation of siRNA oligonucleotides targeting the fusion oncogene TMPRSS2-ERG with squalene and solanesol through the copper-free azide–alkyne cycloaddition (SPAAC) was published (Scheme 24) [75]. The reactive unit dibenzocyclooctyne (DBCO) was attached to the 5′ end of the sense oligonucleotide strand through a long linker to avoid steric hindrance (Scheme 24, compound 94). Azido-squalene **95** and azido-solanesol **96** were clicked in Cu-free optimized conditions (DMSO/H_2_O/acetone mixture; 1:50 alkyne/azide molar ratio) to yield conjugates **97** and **98** with yields up to 95% when the reaction mixture was stirred for 12 and 18 h, respectively, at room temperature and avoiding the formation of side products during the synthesis. The same compounds **97** and **98** were also obtained by annealing the antisense strand to the sense strand after conjugation of squalene and solanesol (data not shown in Scheme 24). The bioconjugates were able to self-assemble in nanoparticles in aqueous solution. Specifically, four nano-formulations, two for **97** and two for **98**, were obtained by annealing the sense strand either before or after the conjugation with terpenoids. Well-defined, spherical and compact objects were obtained when **97** was formed by annealing after conjugation and when **98** was obtained by annealing before conjugation. However, no nano-formulation was able to enter the cells in vitro without the use of cationic polymers. On the other hand, the well-defined nano-objects obtained showed significant tumor growth inhibition by 50–60% in mice bearing VCaP prostate cancer xenografts, suggesting a sensitive correlation to their different biodistribution behavior. Indeed, the nano-formulation of **97** obtained by annealing after conjugation was found to accumulate in the prostate tumor.

The orthogonality of the CuAAC was exemplified in the work of Zewge and co-workers, where a sense RNA, employed for siRNA approach, was internally modified by subsequent amidation and cycloaddition reactions to provide a platform for rapid structure–activity relationship analysis of variously substituted oligonucleotides [76]. An optimized multicomponent orthogonal protocol was developed to obtain various heavily modified siRNAs composed of multiple amide and triazole conjugates. A representative example of this approach is shown in Scheme 25 for the synthesis of heavily conjugated **100**. The chosen modifiers were those targeting sugar GalNAc for the click ligation and the lipophilic pentylamine for the amidation. Complete incorporation was achieved when amidation was performed first, followed by CuAAC reaction with the complex CuBr·SMe_2_ in a DMSO:H_2_O mixture. Interestingly, after lipid-mediated in vitro transfection, modified siRNA retained the ability to reduce ApoB (apolipoprotein B) mRNA levels. Moreover, when GalNAc was present in a clustered manner on the sense strand, binding of siRNA to ASGPR (Asialoglycoprotein Receptor) was improved, proving that heavy modification in the 2′ ribose position of the sense strand is possible in the siRNA therapeutic approach.

Specific antisense oligonucleotides were designed and synthesized as promising candidates as antibacterial drugs. The main obstacle in the development of new oligonucleotide antibiotics comes from the fact that bacterial cells do not spontaneously uptake oligonucleotides from their environment [77]. However, Gryko et al. reported that conjugation to vitamin B_12_ enabled the uptake of oligonucleotides in *Escherichia coli* and *Salmonella enterica* [78]. The stable vitamin B_12_-2′OMeRNA conjugate **103a** targeting the bacterial mRNA encoding RFP (red fluorescent protein) and its scrambled sequence **103b** were prepared by optimized CuAAC reaction in solution phase (Scheme 26). Conjugated **103a** allowed a reduction in the fluoresce levels of the studied bacteria of 50% in respect to the untreated control and to the control treated with the scrambled oligonucleotide **103b**. Furthermore, conjugated **103a** was found to be not toxic on human embryonic kidney cells 293 (HEK 293) after 48 h, thus representing a possible candidate for future development of oligonucleotide-based antibiotics.

Boron clusters have been used in medicinal chemistry predominantly against cancer via boron neutron capture therapy [79]. Recently, a general and convenient approach based on the incorporation of different boron clusters at specific positions of DNA oligonucleotide strands was developed by post-synthetic CuAAC modification (Scheme 27) [80]. The silencing activity of diversely conjugated oligonucleotides was evaluated by silencing of EGFR (epidermal growth factor receptor) mRNA in vitro. The EGFR gene is overexpressed in various human cancers and is considered a good therapeutic target for antisense oligonucleotides [81]. The silencing activity of the oligonucleotides **105a** modified at the nucleobase was found to be lower than that of unmodified oligonucleotide. However, oligonucleotides **107a**–**c** showed silencing levels promising for potential therapeutic applications (83–95% decrease in EGFR expression). The same approach was fruitfully employed for the synthesis of more heavily modified, boron cluster-bearing antisense oligonucleotides [82]. Up to five azide-containing boron clusters were incorporated at the 2′ ribose position of an EGFR targeting sequence post synthetically by CuAAC. The obtained oligonucleotide **110** demonstrated increased lipophilicity and low cytotoxicity (MTT assay on HeLa cells). However, the silencing activity of **110** was found to be lower than that of the control-unmodified oligonucleotide. This could be attributed to a less favorable recognition of the complex RNA/DNA by RNase H due to the boron cluster presence on the antisense strand.

Several synthetic splice-switching oligonucleotides (SSOs) targeting nuclear pre-mRNA molecules have been recognized by FDA as drugs, and others are in clinical trials [83]. However, low cell and in vivo oligonucleotide uptake limited their development for therapeutic applications [84]. To overcome this drawback, a recent work [85] described a new synthetic methodology for conjugation of two morpholino oligonucleotides (PMO) to a single unit of the cell-penetrating peptide Pip6a (Scheme 28). Three different orthogonal conjugations including amide, disulfide and triazole linker chemistries were studied. Interestingly, bi-specific modified SSOs **113a****,b** in which *Dmd* and *Acvr2b* targeting PMO were conjugated via alkyne-azide click chemistry as shown in Scheme 28, were the most active in both genes of a mouse model of Duchenne muscular dystrophy [85].

In a striking example of click reaction use, Murthy and co-workers [86] employed two different azide-alkyne cycloadditions to assemble a multifunctional oligonucleotide, named DART (DNA assembled recombinant transcription factors), capable of delivering transcription factor proteins with high efficiency in vivo. The authors envisaged a modified oligo DNA possessing four galactose units targeting liver cells, linked to two lipophilic chains by an acid-cleavable acetal moiety (Figure 4). This endowed the oligonucleotides with endosome-escaping abilities and resulted in the successful delivery of the transcription factor nuclear erythroid 2-related factor 2 (Nrf2) to the liver, which triggers the expression of downstream anti-inflammatory-related genes. The efficacy of DART was proved by the successful rescue of mice from acetaminophen-induced liver damage. Importantly, the CuAAC reaction permitted the conjugation of the galactose units to the acetal linker, while the SPAAC was employed for the conjugation of the DNA strand to the previously obtained modifiers.

Nucleic acid aptamers are short single-stranded DNA or RNA (ssDNA or ssRNA) molecules and are commonly produced by systematic evolution of ligands by exponential enrichment (SELEX). They can selectively bind to a specific target, including proteins, peptides, carbohydrates, small molecules, etc., with recognized therapeutic applications [87]. In a recent work, a novel method named “click-SELEX” was described for the preparation of nucleobase-modified nucleic acid libraries by using CuAAC chemistry [88]. In brief, “clickmers” were prepared starting from an alkyne-modified DNA library **114** (Scheme 29A) in which conventional thymidine was replaced by C5-ethynyl-2′-deoxyuridine (EdU); then the library was further modified through click conjugation of small molecules, such as the azido-indole **115**. In a typical click-SELEX cycle (Scheme 29B), after incubation with the target molecule, the selected sequences were amplified by PCR in a solution containing the alkyne-modified nucleotide EdU. After digestion of the 5′-phosphorylated antisense strand by exonucleases, the CuAAC was repeated to reintroduce modifications, and then a new selection was carried out. Furthermore, this approach could not be limited to a single modification and could give rapid access to a plethora of novel aptamers modified with molecular entities that are not compatible with solid-phase or enzymatic synthesis conditions, thus satisfying the demand for specifically tailored aptamers. In a more recent paper [89], the specific aptamer Gint.4T targeting the platelet-derived growth factor receptor-β (PDGFRβ), was successfully exploited for the selective delivery of a small therapeutic peptide to cardiac cells (Scheme 30). The conjugation reaction was performed by click chemistry in the presence of *N*,*N*,*N*′,*N*′,*N*″-pentamethyldiethylenetriamine (PMDETA) as a Cu (I)-stabilizing agent. The full restoration of the normal levels of L-type calcium channel, which can recover myocardial function in pathological heart conditions, was achieved.

## 4. Conclusions

In conclusion, this review highlights the extraordinary diversity of pharmacological applications related to the use of one of the most important click reactions, the 1,3-dipolar cycloaddition of alkyne and azide to form 1,2,3-triazole moiety, in the field of nucleosides, nucleotides and nucleic acids. 

Nucleoside, nucleotides and nucleic acids are biomolecules with a crucial role in all biological cellular life functions; thus, the development of analogs for diverse biological targets represents a research topic of great interest. Considering the well-established role of 1,2,3-triazoles as biocompatible pharmacophores and bioisosteric linkers, click reactions are a very attractive tool for the construction of modified and/or conjugated nucleosides, nucleotides and nucleic acids including oligonucleotides. In addition to the high efficiency of these reactions, azide and alkyne moieties can be easily incorporated into oligonucleotides without affecting biophysical and biochemical properties. Azide-alkyne cycloadditions can be carried out in solid-phase conditions and can give rise to a plethora of novel, tailored modified nucleic acids and modified nucleic acid libraries.

Furthermore, the present review indicates that, in recent years, many studies have been dedicated to oligonucleotide modifications using click azide-alkyne cycloaddition. Studies in vitro and in vivo show that several pharmacokinetic and pharmacodynamic properties of modified compounds result improved, suggesting that the 1,2,3-triazole does not act as a passive linker but, on the contrary, is an important moiety in drug design and delivery. 

Finally, although many nucleosides and oligonucleotides possessing the 1,2,3-triazole moiety have been reported in the last years, further studies are required to assess the in vivo efficacy and low toxicity of lead compounds. In the future, we believe that more elucidations about 1,2,3-triazole chemistry, biochemistry and biocompatibility would facilitate the development of triazole-based nucleosides, nucleotides and nucleic acids for clinical applications.

## Data Availability

Not applicable.

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
