# Peer review of "Modified Nucleosides, Nucleotides and Nucleic Acids via Click Azide-Alkyne Cycloaddition for Pharmacological Applications"

_molecules, 2021, doi:10.3390/molecules26113100_

Round 1
Reviewer 1 Report
The authors in this review describe the click reactions for the construction of modified and/or conjugated nucleosides, nucleotides and nucleic acids to generate 1,2,3-triazoles as biocompatible pharmacophores and bioisosteric linkers. As a result, a vast array of compounds have been obtained with a wide range of biological activities. The manuscript can be recommended for publication almost as it is provided the following issued will be addressed:
Line 109-110: the correct name of the compound is tauroursodeoxycholic acid (not ...desoxy...).
Figure 1: the difference between cyclooctene regiosisomers is not discernible in the case of a general formula (panel D).
Steroid structures (Scheme 1): the bond between C17 and C20 cannot be a wedged stereobond (as a rule, two stereocenters cannot be connected by stereobond, cf. Brecher, J. Graphical representation of stereochemical configuration (IUPAC Recommendations 2006). Pure Appl. Chem. 2006, 78 (10), 1897–1970).
Scheme 7, 8: the solid bonds (used for perspecivization) and wedged bonds (used as stereobonds) should not be used simultaneously.
Refs. 31, 49, 53, 73: please check the journal abbreviations.
Ref. 72: check this reference.
Ref. 74: please check the author name.
The authors claim that they focused on articles published since 2015 but they have omitted the paper by Bodnar et al. from 2016 on steroid-nucleoside bioconjugates (Molecules 2016, 21, 1212; doi:10.3390/molecules21091212).
Author Response
We thank the Reviewer for giving us the opportunity to submit a revised draft of the manuscript “Modified Nucleosides, Nucleotides and Nucleic Acids via Click Azide-Alkyne Cycloaddition for Pharmacological Applications” for publication in Molecules. We have appreciated and incorporated all suggestions made by the Reviewer. Please, see the attached file for a point-by-point response.

Reviewer 2 Report
The review by Perrone et al. is a review of the published data on the chemistry of components of nucleic acids (nucleosides, nucleotides and oligonucleotides) based on the Huisgen’s 1,3-dipolar cycloaddition reaction between azide and alkyne groups (click reaction) to obtain nucleic acids analogs and derivatives, potential pharmaceuticals and compounds for the use in biomedical and biophysical research. The text includes an overview of works published in the last few years (since 2015). It is well constructed and divided into four sections (Introduction, review, conclusions and references). In the Introduction the mechanism of Huisgen's 1,3-dipolar cycloaddition reaction under various conditions with or without a catalyst is well described. Examples of the click reaction performed on nucleosides and nucleotides, and then on oligonucleotides (hereinafter referred to as "nucleic acids", although more specifically you could use "oligonucleotides") are discussed. This last section contains citations of triazole modifications within the internucleotide linkage as well as within other fragments of the oligonucleotides. The works using the methodology of post-synthetic modification in the click reaction on the solid support and in solution are discussed separately. The overview includes also a brief conclusion section and a reference list.
In general, the review could be more critical, with more data on the properties of obtained conjugates. The review needs more conclusions about the compatibility of the click reactions in cellular conditions, and triazole moiety in molecules of biological importance (e.g. in comments to Figure 3). It could underline the importance of the click approach in generation of derivatives of pharmaceutical importance (more data given).
Major revisions:
A more detailed discussion of Schemes 6, 8, 9, 10, is needed providing both more details on the individual transformations and on the biological results (e.g., comparison of the antitumor properties of compound 48 and ribavirin, or the antiviral properties of compounds 50)).
Scheme 2 - in the frame of the Molecules journal regulation the authors might mention their unpublished works, or neglect these data. Anyway, Appendix A is not necessary.
Scheme 7, the data presented are in contradiction to the text (IC50 and CC50 values)
Scheme 15 needs reorganization. Compound 62 should be labeled as 62A, B and C and with description below the structure of the R meaning for A, for B and for C. Three arrows from 62 + 63 labeled with "a" above the arrow and "A" below the arrow (and also "a" and "B" and "a" and "C") lead to products 64, 65 and 66.
I propose to add a few sentences about the cited work [49] on the biocompatibility of the triazole ring in the structure of a DNA duplex.
Ref 41 should be more specific.
Scheme 17 - Triazole-linked morpholino analogs should be compared to other models (data cited)
Scheme 18 - short data about in vivo studies of the click products is needed
Fig. 4 - double click-modification should be described more precisely, and data addressing the influence of four triazole rings on the biological properties of the prepared oligonucleotides should be given
Scheme 22 - “two of the four tested nanoformulations” should be specified
Scheme 25 - needs correction - N3-BC numbering is wrong (102 means the same as 100); besides, in original paper these compounds are used for the labeling of both compounds, 97 and 99.
The same authors applied post-synthetic click modifications for introduction of several molecules of 100c, what demonstrates the potential of the click approach for the multiple oligonucleotide modification – worthy to mention
Scheme 26 - needs renumbering - also correct the number of the final products 104 (it should be 104 a-d)
Minor corrections:
- line 100 - "considered an interesting platform" replace for "considered as an interesting"
- line 123 - in schemes caption make unification for a) or b) etc. or (a) or (b). Do not use “: “
- line 130 remove Appendix A,
- Line 132 Scheme 2: correct compound number for 17
- line 186-190 scheme 6 is very unclear, the numbering of compounds is not discussed the the
- line 196 put"(IC50)" next to "lower cytotoxicity"
- line 212 – scheme 8 needs better presentation
- line 218 - remove "non-natural"
- line 224 missing is a bracket after "yield",
- Line 228 - correct “nucleoside”
- Line 238 - “for” imaging
- line 241 - correct “reported”
- line 240 and others - use coma for: ,respectively,
- scheme 11, give the structure of the substrate 4
- line 252 "in respect", "remarkable"
- scheme 12, there is no obvious difference in R substituent, in caption under the reaction arrow
- line 261 - via CuSO4 / ascorbate
- Scheme 13 - explain Ohira-Bestmann reagent
- Scheme 14 - draw the structure for 10 and TMSN3
- reword the sentence line 294 - Oligonucleotides are nucleic acid therapeutics…
- Scheme 16 - in the structure 67 DMTr group should be written DMTrO- as in the products
- line 358 should be "independently"
- Line 368 - significantly
- line 371 should be "exonuclease"
- Scheme 20 – Compound’s number (83) is missing (the product is indicated in the text)
- line 402 no buffer pH is specified, it should be unified in all captions
- line 417 - reference needed for description of the BBB penetrating peptide MIF1
- line 425 and 427, unify : 30min or 0.5h
- lines 453 and 461, unify either overnight or 12h
- line 476 please, define the 2'-position: where? Sugar residues in the oligonucleotide chain
- Scheme 23 – please, verify whether 93 is still on the solid support, or is removed. Make the correction either in the scheme or in the caption
- line 498 -please define the kind of buffer
- 32, line 505 "considered as"
- scheme 25 - check the reaction conditions
- 34, line 515 - give the reference after “a recent work”
- line 562 - give the reference after "In a more recent work"
References:
please, unify the references format, line 614 correct for "Huisgen", cite properly the authors names (e.g. Ref 74)
Author Response
We thank the Reviewer for giving us the opportunity to submit a revised draft of the manuscript “Modified Nucleosides, Nucleotides and Nucleic Acids via Click Azide-Alkyne Cycloaddition for Pharmacological Applications” for publication in Molecules. We have appreciated the time and effort that the Reviewer dedicated to provide a feedback on our manuscript. We are grateful for the comments and suggestions leading to valuable improvements to our manuscript. We have revised the introduction by adding several sentences about 1,2,3 triazole features and biocompatibility. See lines 69-80 in the revised manuscript. Conclusions have also been revised. We have incorporated all suggestions made by the Reviewer. Please, see the attached file, for a point-by-point response.

Reviewer 3 Report
Perrone and coworkers summarized the applications of click reactions for the functionalization of nucleosides, nucleotides and nucleic acids. The review covers a wide range of reactions and is composed of well-organized schemes and figures. It is recommended for publication with minor corrections.
- Line 51, the authors indeed tried to be careful with the "orthogonality" of copper catalyst, yet the sentence would be more straightforward if it starts with "the major drawback of CuAAC is that the Cu(I) catalyst promotes...".
- A short discussion on the biocompatibility of click reaction products, especially on the degradability and toxicity of the triazole linkages.
- The authors are suggested to make a summary table for the ongoing clinical trials involving nucleosides, nucleotides and nucleic acids that are fabricated with click reactions.
- More references on click reactions shall be included in the Introduction: DOI: 10.1002/9780470559277.ch110148; DOI: 10.1002/cpch.85; DOI: 10.1016/j.trechm.2020.03.007.
Author Response
We thank the Reviewer for giving us the opportunity to submit a revised draft of the manuscript “Modified Nucleosides, Nucleotides and Nucleic Acids via Click Azide-Alkyne Cycloaddition for Pharmacological Applications” for publication in Molecules. We have appreciated and incorporated most of the suggestions made by the Reviewer. Please, see the attached file, for a point-by-point response.

Round 2
Reviewer 2 Report
- I would recommend to put hyphen along the text for 1,2,3-triazole, 1,5- and 1,4-triazoles, 1,3-dipolar cycloaddition etc.
- line 132 twentythree – space: twenty three
- Scheme 6 – correct the structure of steroid moiety in compound 31
- line 210: correct for: coupling of the
- Scheme 1-: transformation b and c (from 50 and 51 to 52) is not clear and needs a sentence of chemistry description
- Scheme 14: correct for: Bestmann–Ohira reagent CH3COC(=N2)PO(OCH3)2
- line 309: change …novel nucleosides via Cu-AAC click reaction with antibacterial activity.” for novel nucleosides with antibacterial activity via Cu-AAC click reaction.
- line 311. This chemistry is not clear, and should be corrected to: Subsequent functionalization of hydroxymethyl residue to aldehyde and then to alkyne moiety, followed by the coupling….
- Scheme 27, correction of point 11 from the previous review is not satisfactory – the same authors published multiple post-synthetic click approach obtaining the high boron cluster models for bnct (Kaniowski et al , Molecules 2017). Needs clarification/ reference / figure adding.
- The answer of point 12 of previous review is also controversial, as two 109 (109a and 109b) reacting with two 110 (110a and 110b) give four combinations. However, I guess the original paper presents these two products (D2 and D3), as these are bi-specific. The other two combinations would give specific one-gene specific ( for dmd or for Acvr2b gene).
- line 629 – correct to: DMD
Author Response
We have incorporated most of the suggestions made by the Reviewer. Please, see the attached file, for a point-by-point response.
